# The Risk of Obstetrical Hemorrhage in Placenta Praevia Associated with Coronavirus Infection Antepartum or Intrapartum

**DOI:** 10.3390/medicina58081004

**Published:** 2022-07-27

**Authors:** Irina Pacu, Nikolaos Zygouropoulos, Alina Elena Cristea, Cristina Zaharia, George-Alexandru Rosu, Alexandra Matei, Liana-Tina Bodei, Adrian Neacsu, Cringu Antoniu Ionescu

**Affiliations:** 1Department of Obstetrics and Gynecology—“Carol Davila” University of Medicine and Pharmacy, 050474 Bucharest, Romania; irinapacu@hotmail.com (I.P.); george.rosu@drd.umfcd.ro (G.-A.R.); alexandra.matei@drd.umfcd.ro (A.M.); liana.tina.bodei@rez.umfcd.ro (L.-T.B.); cringu.ionescu@umfcd.ro (C.A.I.); 2“St. Pantelimon” Emergency Clinical Hospital, 021623 Bucharest, Romania; nz@zitaquality.ro (N.Z.); alina.elena.cristea@rez.umfcd.ro (A.E.C.); cristina.zaharia@rez.umfcd.ro (C.Z.); 3Bucur Maternity “Saint Ioan” Clinical Hospital, Strada Bucur nr. 6, 012363 Bucharest, Romania

**Keywords:** obstetrical bleeding, SARS-CoV-2, placenta previa, placenta accreta

## Abstract

*Background and Objectives*: The aim was to evaluate the severity of obstetrical bleeding in the third trimester associated with COVID infection in placenta previa and accreta. *Materials and Methods*: A retrospective study was conducted to compare the risk of obstetrical bleeding in the case of placenta previa with or without associated SARS-CoV-2 infection. Patients presenting with placenta previa before labor were classified into three groups: group A (control) as no infection throughout their pregnancy, group B as confirmed infection during the 1st trimester, and group C as confirmed infection at the time of delivery. Infected patients were stratified according to the severity of signs and symptoms. The severity of obstetrical hemorrhage at birth was assessed quantitatively and qualitatively. All placentas were analyzed histologically to identify similarities. *Results*: Prematurity and pregnancy-induced hypertension appear significantly related to SARS-CoV-2 infection during the 3rd trimester. Placenta accreta risk increases significantly with infection during the 1st trimester. No statistically significant differences in the severity of hemorrhage associated with childbirth in cases with placenta previa between groups A and C but increased obstetrical bleeding mainly due to emergency hemostatic hysterectomy in group B driven by placenta accrete were detected. Obstetrical hemorrhage at birth in the case of coexistence of the infection was found not to correlate with the severity of the viral disease. Meanwhile, the number of days of hospitalization after birth is related to the specific treatment of COVID infection and not related to complications related to birth. *Conclusions*: The study finds an increased incidence of placenta accreta associated with placenta previa in cases where the viral infection occurred in the first trimester of pregnancy, associated with an increased incidence of hemostasis hysterectomies in these patients. Placental histological changes related to viral infection are multiple and more important in patients who had COVID infection in the first trimester.

## 1. Introduction

The COVID-19 pandemic is a reality of the last few years that has created important medical problems that were unforeseen and difficult to control, evaluate, and solve in absolutely all medical fields. Some of the most frequently encountered and controversial pathologies are related to the spectrum of coagulopathies, thrombosis, and bleeding associated with infection with the SARS-CoV-2 virus, with specific complications [1].

The impact of SARS-CoV-2 infection on the pregnant woman, fetus, and newborn is still little known, both regarding the short-term consequences during pregnancy, at the time of birth, or the long-term, distant consequences. The histopathological changes at the placental level during pregnancy are already known: changes that are nonspecific but are responsible for the increased rate of premature birth and hypertension induce pregnancy [2]. Although the syncytiotrophoblast is an effective barrier to viral infections, the existence of infection at the placental level and transplacental transmission to the fetus during pregnancy by aspiration of amniotic fluid has been demonstrated [2,3,4].

Numerous clinical data suggest that exacerbation of the immune response in pregnancy associated with cytokine storm caused by infection with the SARS-CoV-2 virus could more easily trigger varying degrees of consumption coagulopathy [5,6]. There are currently insufficient data on the severity of obstetrical bleeding in the third trimester associated with COVID infection.

In recent years, the incidence of hemorrhages in the third trimester is increasing especially because of the increased incidence of births by cesarean section in the context of practicing defensive obstetrics, the increasing maternal age at first birth, and the use of assisted human reproduction techniques [7]. We can already observe the negative consequences related to this aspect by the increasing number of pregnant women with non-intact uteri (due to prior delivery by cesarean section) which also leads to an increased incidence of low-lying placenta and abnormal adhesions to the placenta in the following pregnancies. We can currently say that placenta previa is the main cause of obstetrical bleeding in the third trimester of pregnancy [8].

## 2. Materials and Methods

We conducted a retrospective study whose objective is to compare the risk of obstetrical bleeding in the case of placenta previa associated with SARS-CoV-2 infection with the risk of obstetrical bleeding related to the placenta previa unrelated to the SARS-CoV-2 infection. The study included patients who gave birth in the St. Pantelimon Obstetrics and Gynecology Clinic in Bucharest and Bucur Maternity of Saint Johns’ Hospital in Bucharest between January 2018 and December 2021. During the 2020–2022 period, Bucur Maternity acted as a support maternity unit for pregnant women with SARS-CoV-2 infection.

The study was carried out with the agreement of the Ethics Council of the Hospital of Obstetrics and Gynecology “Saint Pantelimon”, Bucharest, and of the Ethics Committee of the Bucur Maternity “Saint John’s” Clinical Hospital, Bucharest.

Three study groups were considered:Group A—patients with placenta previa who do not have SARS-CoV-2 infection diagnosed from the moment of conception until birth;Group B—patients with placenta previa who have a history of SARS-CoV-2 infection in the first trimester of pregnancy and who are negative at the time of birth;Group C—positive SARS-CoV-2 patients at the time of birth only.

Patients that were diagnosed positive for SARS-CoV-2 at the time of birth and were also positive in the first trimester and/or second trimester of pregnancy were decided to be excluded from the study to isolate the effects of infection per trimester. No patients were found to fit in the above-mentioned situation. Similarly, in the case that patients were found to be positive during the second trimester but not during delivery, it was decided that they should form a group of their own. Again, no such patients were identified.

The demographic, obstetrical, birth method, and status/history of SARS-CoV-2 infection were obtained from the patients’ observation sheet. Positive SARS-CoV-2 infection was diagnosed based on the results of the nasopharyngeal sample collected at the time of hospital admission and a positive result validated by an accredited laboratory; either PCR tests for the SARS-CoV-2 virus or antigenic tests were considered. Only patients for whom there was a positive PCR test result or an antigen test performed at an accredited laboratory during the first or second trimesters were considered as having reliable evidence of past infections.

SARS-CoV-2 infection at the time of birth was classified in the following categories [1,9]:Asymptomatic if the PCR test is positive but without respiratory or general symptoms;Mild if there were any of the following signs or symptoms: fever, chills, mild cough, headache, etc., but without shortness of breath, chest pain, or breathlessness;Moderate if there are respiratory difficulties, suggestive pulmonary imaging, and/or SpO_2_ > 94%;Severe if the respiratory rate is over 30 breaths per minute, SpO_2_ < 94%, severe breathlessness, cough, altered general condition, and severe respiratory failure.

The severity of obstetrical hemorrhage at birth was assessed according to the preoperative and postoperative values of hemoglobin and the calculated percentage loss of blood volume taking into consideration the effect of the number of blood units (erythrocyte concentrates) that were transfused. The average duration of the days of hospitalization and the intra- and post-operative complications were used as a qualitative aspect of the severity of obstetrical hemorrhage according to the data from the specialized literature [1,10,11]. All these aspects were retrospectively recorded from the patients’ observation sheets. 

The diagnosis of placenta previa was made based on the preoperative ultrasound examination performed during the evolution of pregnancy after 20 weeks of amenorrhea, with the classification of the anatomical variants of placenta previa being made as follows [12,13]:Grade I: low-lying placenta: placenta lies in the lower uterine segment, but its lower edge does not abut the internal cervical orifice (lower edge 0.5–2.0 cm from the internal orifice);Grade II: marginal previa: placental tissue reaches the margin of the internal cervical orifice, but does not cover it;Grade III: partial previa: placenta partially covers the internal cervical orifice;Grade IV: complete previa: placenta completely covers the internal cervical orifice.

The groups were compared with descriptive and bivariate statistics using Student’s *t*-test for continuous variables. Chi-square test or Fisher’s test were used for categorical variables. ANOVA test was used to determine if there is a statistically significant difference between two or more categorical groups. All analyses were completed using Addinsoft (2022) and XLSTAT statistical and data analysis solution, New York, NY, USA.

We analyzed macroscopic and histopathological aspects of all the placentas with the help of the Department of Pathological Anatomy within the two clinics. Ultrasound examination was performed before birth, and the diagnosis of placenta accreta was suspected in all cases. The definite diagnosis of placenta accreta was made only after the histopathological evaluation of postoperative certainty. In all cases with placenta accreta, an emergency hemostatic hysterectomy was performed.

## 3. Results

Between January 2018 and December 2022, the number of births in the two clinics was 10,026. The study group includes 154 patients who gave birth in these two maternity wards between January 2018 and December 2021, namely 87 (56.49%) cases during 2018–2020 and 67 (43.51%) cases during 2020–2021 with the diagnosis of placenta previa.

Group A includes 120 negative cases during pregnancy and childbirth for COVID infection (77.92%), group B includes 15 cases that presented the infection in the first trimester (9.74%), and group C includes 19 positive cases for COVID at birth (12.34%).

Demographic data and aspects related to SARS-CoV-2 infection are presented in Table 1. *p*-values shown in the tables are derived using one-way ANOVA testing between all three different groups unless otherwise stated, where groups were tested in pairs using *t*-test. In some cases, *p*-value was not calculated on the basis that either it was not significant, or comparisons between the related groups were considered either out of the scope of this research or not clinically significant.

Table 2 includes comparative data on the severity of obstetrical hemorrhage between the three study groups.

We also performed a comparative analysis between the severity of obstetrical hemorrhage and the clinical form of the COVID infection, with data presented in Table 3.

Table 4 presents the analysis of the causes of hysterectomy of hemostasis in the study groups.

From the histopathological point of view, placental changes were analyzed in all groups, comparatively analyzing (Table 5) the incidence of abnormalities associated with COVID infection in pregnancy, according to the studies carried out so far [2,4].

## 4. Discussion

Regarding the demographic data, there are no statistically significant differences in terms of age, parity, BMI, or tobacco consumption between group A and group B or C as proven when testing for homogeneity.

There are no statistically significant differences between the obstetrical data, but still, the patients who had COVID in the first or third trimester during pregnancy have an increased rate of premature birth, independent of the coexistence of the placenta previa. This increase premature birth is only found to be statistically significant for group C (*p* = 0.0345, < 0.05) and not for group B (*p* = 0.0675, < 0.05). These data are similar to those in the literature that attests to the increased incidence of premature birth and the association with pregnancy-induced hypertension in the 3rd trimester (17% vs. 8.7% for preterm birth, 18.9% vs. 7.8% for preeclampsia) [1,14,15]. When analyzing using unadjusted odds ratios, it was found that it is 45.4% for group C and only 12.4% for group B more likely in comparison to group A to deliver prematurely.

Concerning the fetal outcome, as defined by fetal weight and Apgar score, homogeneity was confirmed between group A and group B or C. There are no statistically significant differences in fetal prognosis (weight, Apgar score). The birth was performed in 125 cases (81.17%) by cesarean section and in only 29 cases (18.83%) vaginally, with these being the cases with minimal bleeding and 1st degree placenta previa. There are no significant differences in the mode of delivery between groups, and correlation testing proved that COVID infection, albeit in the 1st trimester or active irrespective of severity, did not significantly influence delivery mode.

There are no statistically significant differences (*p* >> 0.05 using one-way ANOVA) between the three groups regarding the type of placenta previa.

An increased incidence of placenta accreta is observed in the group of patients who had SARS-CoV-2 infection in the first trimester of pregnancy (weeks 5–14 of amenorrhea) at the time of trophoblast formation. The odds ratio (OR) for placentation not to be accreta in group B was calculated at 0.22 (0.065–0.759, CI 95%) and 0.592 in group C (0.151–2.332, CI 95%).

Most COVID infections are asymptomatic and mild, without statistically significant differences between the three groups. There was only one serious case with respiratory failure that required hospitalization in the intensive care unit, with the birth being performed by emergency cesarean section surgery for maternal purposes at 35 weeks of amenorrhea. Maternal and fetal evolution was favorable, with an increased number of days of hospitalizations (27 days).

It is worth noting an increased incidence of pregnancies with placenta previa (154 cases out of the total of 10,026 births, 1.536%) compared to the data from the specialized literature, (0.15–1.1%) [16,17], which is explained by the framing of the two centers in which the study was conducted in the category of grade III maternity wards, with a significantly increased addressability of pregnancies with high obstetrical risk. In addition, between 2020–2021, Bucur Maternity Hospital was the only unit in Bucharest and the surrounding areas dedicated to childbirth assistance exclusively for patients with COVID infection. There are no differences in the incidence of pregnancies with placenta previa in the period 2018–2019 (1.5%) compared to the period 2020–2012 (1.582%).

To date, there are very few studies to try to evaluate blood loss at birth and postoperative morbidity in obstetrical bleeding associated with COVID infection, and the studies that have been conducted include a small number of cases [1]. For the non-pregnant population, there are studies and case presentations that report morbidity and mortality associated with COVID infection, and the data are still controversial due to the heterogeneity of laboratory data and the difficulty of unitary quantification of blood loss. More data are needed in this regard, and the present study aims to guide the public and anesthesiologists in establishing as correctly as possible the hemorrhagic risk for these pregnant women, especially when it is associated with other obstetrical causes of severe hemorrhage.

### 4.1. Hemostasis Changes in COVID Infection

Abnormalities of blood clotting are characteristic of COVID infection, directly proportional to the severity of the disease. Starting from an altered inflammatory response, the balance between pro and anticoagulant factors changes and endothelial dysfunction plays a major role [18]. Changes in blood growth during the COVID infection proved to be the main prognostic factor regarding the evolution of the disease [19].

There is also thrombocyte hyperactivity that is more pronounced in severe cases and that participates in thrombotic complications [18]. Patients with increased platelet/lymphocyte ratio have an increased duration of the disease [19,20]. The SARS-CoV-2 virus inhibits hemopoiesis via CD-13 receptors, which leads to the possibility of thrombocytopenia, especially in the case of increased bleeding in other coexisting pathologies. [18]

The initial procoagulant status was initially considered to be part of the first phase of the disseminated intravascular coagulation (DIC) process but is currently considered not to meet the DIC criteria of the International Society of Thrombosis and Haemostasis [18].

### 4.2. Hemostasis in Pregnancy

Pregnancy is a procoagulant diathesis with the increase of prothrombotic factors VII, VIII, X, XI, von Willebrand, and fibrinogen and the decrease of the S protein and the alteration of the fibrinolysis. It is mandatory to recognize all prothrombotic risk factors for pregnant women to establish thromboprophylaxis in pregnancy and postpartum [21], with COVID infection being part of this category.

### 4.3. Impact of COVID on Blood Clotting during Pregnancy

Since the first trimester of pregnancy, COVID association increases thrombotic risk, with the risk being maximum in the third trimester, in association with maternal obesity, thrombotic history, thrombophilia, age over 35 years, and diabetes mellitus [3]. The risk is directly proportional to the severity of the viral infection, with the condition of bed immobilization of the pregnant woman adding to the additional factors related to prolonged venous stasis [19,20].

Under the conditions of association of severe obstetrical bleeding in the third trimester with COVID infection, all these factors cause an increased risk of developing DIC [18,19,20].

In our study, initial preoperative/prelabor hemoglobin levels had no significance difference between the three groups (*p* >> 0.05), while postoperative/postpartum hemoglobin levels were significantly different between group A and B (*p* = 0.022) and not between A and C (*p* = 0.0045). Similarly, when blood loss during delivery was expressed as a percentage decrease of hemoglobin during labor, groups A and B had a significant difference (*p* = 0.023), and groups A and C did not (*p* = 0.065). All in all, values of paraclinical analyses and postoperative morbidity did not yield statistically significant differences in the severity of hemorrhage associated with childbirth in cases with placenta previa between groups A and C but did between groups A and B, primarily driven by an increased incidence of obstetrical bleeding and emergency hemostatic hysterectomy in group B of patients who had SARS-CoV-2 infection in the first trimester (*p* = 0.0243, CI 95%). This result is related to a significantly increased incidence (*p* = 0.496, CI 95%) of abnormal adhesions of the placenta (Table 1) and an increased rate of hysterectomies of hemostasis (*p* < 0.05), as shown in Table 2.

General anesthesia by orotracheal intubation (IOT) predominates in the group of patients with placenta accreta (Table 1), but this is mainly driven by operative considerations rather than COVID status.

The results overlap with the data of other studies [1,21] that do not identify an increased risk of obstetrical hemorrhage at birth in the case of coexistence of the infection, and there is no correlation with the severity of the viral disease (Table 3). One-way ANOVA proved no significant differences between groups.

The number of days of hospitalization after birth is increased for medium and severe cases related to the specific treatment of COVID infection and not related to complications related to placenta previa. A correlation study between hospitalization days and infection severity yielded an r^2^ value of 0.886 in comparison to hospitalization days and complications directly linked to PAS of 0.456.

Intraoperative lesions of the bladder, coagulopathy, and reinterventions were more common in group B and related to hysterectomies of hemostasis, but again, statistically, no significance could be proven (*p* >> 0.05). There are no statistically significant differences between uterine relaxation, postoperative infection, or thrombotic complications. It should be mentioned that in all cases, postoperative antithrombotic prophylaxis with low-molecular-weight heparin preparations was performed.

According to the data from the literature [1,3,5], the incidence of emergency hemostatic hysterectomies in pregnancies associated with the placenta previa is increased (4, 4–5, 9%). Further, a statistically significant increase in the incidence of emergency hysterectomies was found between group A and groups B and C cumulatively (*p* = 0.0126, CI 95%), while no significant difference was found between groups B and C.

Table 4 presents the analysis of the causes of hysterectomy of hemostasis in the study groups. It provides the insight that the increased incidence of hysterectomies is mainly driven by PAS in group B. with a significant difference in comparison to groups A and C (*p* = 0.0243, *p* < 0.05, CI 95%), while in group C, there is not a specific cause predominating, and in group A, PAS and uterine relaxation remain the predominant causes.

The necessity of hemostasis hysterectomy was achieved mainly for abnormal adhesions of the placenta (placenta accreta), their incidence being superior in the group of patients who had SARS-CoV-2 infection in the first trimester of pregnancy (*p* = 0.243, a = 0.05), without there being statistically significant differences between the main risk factors of the placenta accreta between the three groups (Table 4). The highest incidence of placenta accreta associated with placenta previa was found in group B of the study, between groups A and C. The OR for placentation not to be accreta in group B was calculated at 0.22 (0.065–0.759, CI 95%) and 0.592 in group C (0.151–2.332, CI 95%). There is an increased risk of placental changes for group B of the patients who had COVID infection in the first trimester of pregnancy compared to those who were positive at the time of birth, i.e., those who presented the infection in the third trimester.

However, analyzing all the cases with placenta previa, we identified that study group B of the patients with a history of SARS-CoV-2 infection in the first trimester of pregnancy and who, compared to the positive patients in the third trimester or those who have no history of COVID infection, had a statistically significant difference regarding the possibility of a severe obstetrical hemorrhage at birth. No testing for COVID-19 has been done at the placental level under any circumstances. No case of COVID infection was registered in newborns in the case of maternal infection at the time of birth; all newborns with COVID-positive mothers were tested at the time of birth. Group B recorded the highest loss of blood volume, on average 17.1 ± 3.5%, followed by group C with 13.5 ± 3.9% in comparison to group A at 11.2 ± 3.5%, with statistical significance when comparing group A with B or C (*p* = 0.0234 and *p* = 0.0445, respectively, for CI 95%) and having considered transfusions. In terms of transfusions, group B had the largest average with 3.9 ± 2.2 units in comparison to group A or B with 2.3 ± 1.1 or 2.1 ± 1.2, respectively. Statistical significance was identified only between groups B and A but not for groups A and C. The average number of days of hospitalization was maximum, where most intra- and postoperative complications occurred. In this regard, it is worth noting the statistically significant difference regarding the hysterectomies of hemostasis, the maximum percentage being in the group of patients who had COVID infection in the first trimester, with most of them having as an indication the placenta accreta. The increased incidence of hysterectomies is mainly driven by PAS in group B with a significant difference in comparison to groups A and C (*p* = 0.0243, *p* < 0.05, CI 95%), while in group C, there is not a specific cause predominating, and in group A, PAS and uterine relaxation remain the predominant causes.

There are no statistically significant differences between the three groups related to the risk factors known for PAS.

Starting from these statistical results, we analyzed data from the specialized literature related to the placental histopathological changes in the COVID infection that could explain the practical clinical aspects revealed by the present study [22,23,24].

It is well-known that the immune status in pregnancy is modified in the sense of adapting to accept an allograft that is the product of conception, leading to a specific immune maternal response to the different types of infections that may occur during pregnancy [25,26]. During the COVID infection, the functionality of the NK cells is also altered, which overlaps with the reduction of their number that occurs physiologically during pregnancy [25,26]. In pregnancy, there are alterations of the CD4+ population, predominantly of the phenotype T-Helper-2 compared to T-Helper-1, which explains maternal immune tolerance [25].

In the task, there is a special pattern of recognition of Toll-like receptors (TLRs), especially TLR4. Three levels of activation of these receptors shall be described. The first level is represented by the activation from the first trimester at the time of implantation of the blastocyst, the second occurs in the second trimester to reduce the pro-inflammatory phenomena that would be physiologically stimulated by fetal growth, and the last is represented by the increased activation in the third trimester to support labor and delivery [26,27]. Infection with the SARS-CoV-2 virus prevents the release of proteins that become ligands for TLR molecules, which exacerbates the immune response from the moment of implantation and formation of the placenta. Further studies will determine whether these phenomena are the basis of the increased susceptibility to infection of the pregnant woman or are protective against infection during pregnancy [26].

The SARS-CoV-2 virus penetrates the nasal mucosa by binding to the angiotensin-converting enzyme 2 receptors (ACE2), which is also found in the digestive tract, placenta, ovary, uterus, and vagina. Intracellular penetration is facilitated by the spike protein at the viral level via trans-serine membranes protease 2 (TMPRSS2) [26]. The cells with the highest susceptibility to being virally infected are those that express both ACE2 and TMPRSS2 [26]. The cells of syncytio- and cyto-trophoblasts express ACE2 from week 7, and thus, there is the possibility of transplacental transmission from the onset of pregnancy.

There are no studies that identify placental pathological changes at the macroscopic level. From the microscopic point of view, non-specific changes such as old or recent microthrombi, infarcts, and fibrin deposits that move the villi of the spines towards the periphery have been described [27]. In cases with reduced maternal-traumatic infusion, the characteristic lesions were those of chorangiosis (the presence of over 10 terminal villi with over 10 capillaries in over three different placental areas), delay in villous maturation [27], and increased syncytial knotting [28]. Chorangiosis is an adaptive response to reduce blood infusion that causes hypoxia and is associated in most cases with maternal desaturation [23]. Increased syncytial knotting appears to be related to the state of hypercoagulability in the COVID infection [29].

Inflammatory lesions at the placental level have been suggested by the increased number of Hoffbauer cells at this level, which is known to be involved in the vertical transmission of viral infections [28]. These are macrophages localized since the beginning of pregnancy at the level of the chorionic villi, being involved in phagocytosis of the apoptotic material and antigen presentation in response to infectious agents.

The starting point of PAS is an incorrect decidualization, which leads to an abnormal invasion of the trophoblast from the moment of its formation. From the histopathological point of view, there is an increased incidence of placental basal chorionic inflammation, maternal vascular changes that lead to abnormalities of maternal circulation, and placental intervillous hemorrhages. Microscopic examination of the placenta will reveal in all cases the presence at the level of the placental basal plaque of myometrial fibers [28].

The development of the placenta is a multifactorial complex and incompletely known process. Hypotheses related to placental abnormal adhesions converge on a combination of the absence of basal plaque, exaggerated extravillous trophoblastic invasion, and abnormal maternal vascular proliferation [24,26,28]. Multiple parallels were made between the abnormal placental invasion in PAS and the tumor proliferation model, in both cases being an increased ability of trophoblastic cells to overcome the local immune systems, induce exaggerated angiogenesis, and activate tissue invasion, often associating a localized inflammatory process [28].

A long-studied aspect is the association between PAS and chronic basal inflammation. It is not known exactly which lymphocytic subpopulations [22] are involved and what their contributions are. Ernst et al. [24] demonstrated that there is an increased lymphocytic infiltrate at the level of the placental implantation site in the case of PAS compared to other pathologies. There appears to be a low number of CD4+ T lymphocytes and an increase in CD25+ lymphocytes compared to normal pregnancy, suggesting a suppression of the immune response mediated by T cells. Increased trophoblastic invasion without differences in cellular dendritic density suggests an immunological dysfunction at the deciduous level [1,22]. In addition, natural killer deciduous cells (dNK), which are the only natural killer cells that play an important role in the early stages of pregnancy, are significantly reduced in the placentas of PAS [22].

There is an obvious overlap between the local and general uterine disturbances that lead to the appearance of PAS in general and the placental changes generated by the COVID infection, which could explain the increased incidence of PAS in cases where the viral infection occurred in the first trimester of pregnancy.

The histological analysis of the placenta previa of the patients included in the present study identifies in the case of patients with infection in the I and II trimesters of the increased prevalence of microcalcifications, fibrin deposits, chorioangiosis lesions, syncial knotting, and local villous inflammation towards patients who did not have the infection in pregnancy. These lesions have significantly increased incidence for patients who had the infection in the first trimester of pregnancy and who present PAS; the group of patients with COVID infection at birth has an increased incidence of local thrombosis. These changes could be in the context of the procoagulant status given by the viral infection or represent a pattern of abnormal proliferation of the trophoblast.

The study makes an important contribution to placental histopathological changes related to SARS-CoV-2 virus infection and the increased risk for abnormal placenta adhesions even in conditions of mild or symptomatic infections. Cesarian is important to be performed in a third-degree maternity ward with addressability for high obstetrical risk cases, including SARS-CoV-2 infection.

The limit of the study is given by the fact that there may be cases of asymptomatic COVID infections, so the number of patients who presented the infection in the first trimester can be changed. These preliminary data should be confirmed by more extensive studies that include a larger number of patients, but the present results may be a starting point for future research.

## 5. Conclusions

SARS-CoV-2 is the pathogen responsible for the current pandemic situation, and there is still no complete data on all the pathophysiological aspects related to this infection. Its implications on pregnancy and maternal and fetal prognosis in the long term is further researched, as the complete discovery of these can be useful in the correct evaluation of many infectious pathologies in pregnancy. There are certain data related to the increased incidence of hypertension induced by pregnancy, premature birth, and low fetal birth weight in pregnant women with COVID infection in pregnancy. There are not yet enough, and complete studies related to hemorrhages in pregnancy associated with this infection and this study can complete these data. The results of the study confirm the data from the specialized literature that do not report an increased incidence and severity of bleeding in the third trimester under the conditions of COVID infection at birth in patients with placenta previa. However, the study finds an increased incidence of PAS associated with placenta previa in cases where the viral infection occurred in the first trimester of pregnancy, associated with an increased incidence of hemostasis hysterectomies in these patients. Consequently, we consider that a perspective study should be pursued regarding timeliness of COVID infection during pregnancy and PAS-associated placenta previa. Further, we wish to draw attention to medical professionals that timeliness of past COVID infections is another parameter that perhaps needs to be taken into consideration while probing for a patients personal history. Lastly, placental histological changes related to viral infection are multiple and more important in patients who had COVID infection in the first trimester.

## Figures and Tables

**Table 1 medicina-58-01004-t001:** Demographic, obstetrical, and related characteristics of SARS-CoV-2 infection. Percentages quoted are based on specific categories expressed in comparison to the respective group.

	Group A COVID-Negative (*n* = 120)	Group B COVID-Positive in Trimester I (*n* = 15)	Group C COVID-Positive during Labor (*n* = 19)	*p*-Value (a = 0.05)
Age (in years)	25.7 ± 5.9	27.2 ± 6.0	25.6 ± 5.7	0.680
Body mass index	24.5 ± 6.1	25.9 ± 7.2	25.1 ± 5.7	0.679
Tobacco consumption				0.785
None Smoker	85 (70.83%) 35 (29.17%)	11 (73.3%) 4 (26.7%)	13 (68.42%) 6 (31.58%)	
Parity				0.885
Nulliparous Multiparous	73 (60.83%) 47 (39.17%)	9 (60.00%) 6 (40.00%)	12 (63.16%) 7 (36.84%)	
Gestational age (in weeks) at the time of delivery				Between: Group A and B: 0.0675 Group A and C: 0.0345
Preterm, < 37 weeks At term ≥ 37 weeks	29 (24.16%) 91 (75.84%)	4 (26.67%) 11 (73.33%)	7 (36.84%) 12 (63.15%)	
Placenta previa				*p* >> 0.05
Grade I Grade II Grade III Grade IV	23 (19.10%) 41 (34.17%) 32 (26.67%) 24 (20.06%)	3 (20.00%) 3 (20.00%) 4 (26.67%) 5 (33.33%)	3 (15.79%) 5 (26.32%) 6 (31.57%) 5 (26.32%)	
Placenta accreta spectrum (PAS)	12 (10.00%)	5 (33.33%)	3 (15.79%)	0.496
Fetal weight (g)	2759 ± 352	2899 ± 458	2650 ± 424	
Apgar Score at 1 min postpartum	8 ± 1	9 ± 1	8 ± 1	
Mode of delivery				>>0.05
Vaginal Cesarean section	23 (19.17%) 97 (80.83%)	3 (20.00%) 12 (80.00%)	3 (15.79%) 16 (84.21%)	
Anesthesia type				
None Spinal/epidural General	12 (10.00%) 86 (71.67%) 22 (18.33%)	2 (13.33%) 10 (66.66%) 3 (20.00%)	2 (10.52%) 10 (52.63%) 7 (36.84%)	
The severity of SARS-CoV-2 infection				
Asymptomatic Mild Moderate Severe	0 0 0 0	3 (20.00%) 7 (46.67%) 5 (33.33%) 0	4 (21.05%) 9 (47.37%) 5 (26.31%) 1 (5.27%)	

**Table 2 medicina-58-01004-t002:** Comparison between the three groups on the severity of obstetrical hemorrhage and postoperative morbidity.

	Group A (*n* = 120)	Group B (*n* = 15)	Group C (*n* = 19)	*p*
Preoperative/Prelabor hemoglobin (g/dL)	11.3 ± 0.7	11.7 ± 1.2	11.5 ± 0.9	>>0.05
Postoperative/Postpartum hemoglobin (g/dL)	10.2 ± 0.6	9.7 ± 1.1	10.2 ± 0.8	Between: Group A and B: 0.022 Group A and C: >>0.05
Decrease in hemoglobin (%)	11.2 ± 3.5	17.8 ± 4.1	13.5 ± 3.9	Between: Group A and B: 0.023 Group A and C: 0.065
Blood transfusion— erythrocytes concentrate (IU)	2.3 ± 1.1	3.9 ± 2.2	2.1 ± 1.2	
Hospitalization (days)				
3–5 days 6–7 days >7 days	45 (37.5%) 59 (49.16%) 16 (13.30%)	3 (20.00%) 7 (46.67%) 5 (33.33%)	5 (26.31%) 10 (52.63%) 4 (21.05%)	
No complications	87 (72.5%)			
Complications				
Bladder injury Coagulopathy Uterine relaxation Infection Hysterectomy Re-exploration Thrombosis	5 (4.17%) 1 (0.83%) 5 (4.17%) 0 12 (10%) 1 (0.83%) 0	2 (13.3%) 1 (6.66%) 0 1 (6.66%) 4 (26.67%) 0 1 (6.66%)	1 (5.26%) 0 1 (5.26%) 1 (5.26%) 3 (15.79%) 0 1 (5.26%)	

**Table 3 medicina-58-01004-t003:** Comparison between the severity of obstetrical bleeding and the clinical form of SARS-CoV-2 infection.

	Asymptomatic Form (*n* = 7)	Mild Form (*n* = 16)	Moderate Form (*n* = 8)	Severe Form (*n* = 1)	*p*-Value (a = 0.05)
Preoperative/Prelabor hemoglobin (g/dL)	11.2 ± 1.2	11.4 ± 1.1	10.5 ± 2.1	9.8 ± 1.5	
Postoperative/Prelabor hemoglobin (g/dL)	10.1 ± 0.9	10.7 ± 1.5	9.8 ± 1.8	9.7 ± 1.2	
Decrease in hemoglobin (%)	10.1 ± 0.5	9.5 ± 1.7	11.5 ± 1.6	6.5 ± 1.9	*p* > 0.05
Blood transfusion— erythrocytes concentrate (IU)	1.6 ± 1.0	1.8 ± 1.6	2.2 ± 2.0	2.1 ± 1.8	
Hospitalization (days)					
3–5 days 6–7 days >7 days	5 (71.42%) 1 (14.29%) 1 (14.29%)	10 (62.50%) 3 (18.75%) 3 (18.75%)	5 (62.50%) 1 (12.50%) 2 (25.00%)	0 0 1 (100.00%)	
No complications					
Complications					
Bladder injury Coagulopathy Uterine relaxation Infection Hysterectomy Re-exploration Thrombosis	0 0 1 (14.2%) 0 1 0 0	1 (6.25%) 0 0 1 3 0 0	1 (12.50%) 0 1 (12.50%) 0 2 (25.00%) 0 1 (12.50%)	1 (100.00%) 1 (100.00%) 0 1 (100.00%) 0 0 1 (100.00%)	

**Table 4 medicina-58-01004-t004:** Causes of hysterectomy of hemostasis per group. Percentages quoted to indicate the percentage of hysterectomy cases per group.

Cause	Group A (*n* = 120)	Group B (*n* = 15)	Group C (*n* = 19)	*p*-value (a = 0.05)
PAS	3 (2.50%)	3 (20.00%)	1 (5.26%)	0.0243
Uterine relaxation	4 (3.33%)	0	1 (5.26%)	
Uterine myoma	2 (1.67%)	1 (6.67%)	0	
Other	1 (0.83%)	0	1 (5.26%)	
Total per group	10 (8.33%)	4 (26.67%)	3 (20.0%)	0.0126

**Table 5 medicina-58-01004-t005:** Comparison of placenta microscopic features of COVID-19 versus the control group A (non-COVID placenta previa group).

Microscopic Features	Group A (*n* = 120), Control Group	Group B (*n* = 15)	Group C (*n* = 19)
Increased microcalcifications	15 (12.50%)	5 (33.33%)	5 (26.31%)
Chorangiosis	5 (4.17%)	4 (26.66%)	3 (15.79%)
Villous agglutination	3 (2.50%)	3 (20.00%)	3 (15.79%)
Increased fibrin deposits	11 (9.17%)	4 (26.67%)	5 (26.31%)
Local thrombosis	13 (10.83%)	5 (33.33%)	4 (21.05%)
Increased syncytial knotting	2 (1.67%)	5 (33.33%)	2 (10.53%)
Delayed villous maturity	4 (3.33%)	3 (20.00%)	2 (10.53%)

## Data Availability

All data (except actual experimental data) and information were collected through open or paid databases containing published journals. Experimental raw data collected during the study are available on request in accordance with the provisions stipulated in the patient consent for publication.

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
