# Peer review of "The Risk of Obstetrical Hemorrhage in Placenta Praevia Associated with Coronavirus Infection Antepartum or Intrapartum"

_medicina, 2022, doi:10.3390/medicina58081004_

Round 1

Reviewer 1 Report

Placenta accreta is a challenging diagnosis under any circumstances. Delivering the pregnant women with placenta accreta is a medicine of a highest quality. But delivery PAS during the pandemic COVID-19 era even more challenging. From one hand its a high risk of bleeding due to placenta accreta and from the other hand its a high risk of thrombosis due to COVID-19. Recent findings showed that SARS-CoV-2 attaches to the ACE-2 receptor that is expressed in the human endometrium and can lead to defective decidualization and abnormal throphoblast invasion that potentially can be implicated in the development of placenta accreta spectrum. Pregnant women with a history of COVID-19 during early pregnancy should be carefully examined and managed. More studies are needed to investigate this aspect of covid-19 pregnancy. From my point of view, authors did a very interesting work that can rise an important questions for further studies.

Author Response

Dear Reviewer,
Thank you for your effort and time spent reviewing our manuscript. We greatly appreciate it.
Indeed, as you mention, recent findings showed that SARS-CoV-2 attaches to the ACE-2 receptor that is expressed in the human endometrium. We do suspect that an active infection during decidualization and trophoblast invasion may be implicated in the development of the placenta accreta spectrum. Hence, we embarked on first try to gain evidence that PAS is correlated to the timing of an active infection during pregnancy.
We look forward to seeing further research done in this field in the future.
Once again, thank you for your constructive feedback.
On behalf of all the authors,
Nikos Zygouropoulos

Reviewer 2 Report

I have read the article “The risk of obstetrical hemorrhage in placenta praevia associated with coronavirus infection antepartum or intrapartum”. Submitted to Medicine.

The aim of this study was to compare the risk of obstetrical bleeding in the case of placenta praevia associated with SARS-COV-2 infection with the risk of obstetrical bleeding related to the placenta praevia unrelated to the SARS-COV-2 infection. 

Although it is a current problem, the increase in hemorrhage due to placenta previa and SARS-CoV-2 infection is not substantiated.

The methodology and the reporting of the results have no order, there are even results (in the tables) that do not have statistical validation (p-value), for example in table 2 Preoperative haemoglobin (g/dl) (there is a difference between the groups?), postoperative Haemoglobin (g/dl) (there is a difference between the groups?). Also the outcomes are not described, which is a serious methodological error, I think the main outcome is bleeding, but secondary? could it well be a composite of adverse maternal or perinatal outcome?

By having the outcome well defined, the results are aimed at solving the problem.

Therefore, the structures of the paper must be reconsidered, since as it currently stands, it is not possible to give it clinical importance.

Author Response

Dear Reviewer,

Thank you for your effort and time spent reviewing our manuscript. We greatly appreciate your insights and have tried to improve our manuscript in line with your recommendations.

As you will notice on some occasions in an attempt to bridge the differences between reviewers’ feedback we were forced to pursue a strategy of "the middle ground between reviewers' suggestions" and not follow entirely upon one's recommendations. Nonetheless, we trust that we have managed to address the most important relevant issues raised given the retrospective nature of this research.

Regarding our design, the initial aim of this study was to identify -retrospectively- if an active infection of SARS-CoV-2 during labour would result in an increase of hemorrhage in cases of placenta previa. During the study, no such association was identified as you mention. An association of increased hemorrhage was identified only in cases when mothers were infected with SARS-CoV-2 during placental formation (1st trimester) correlating with PAS placenta praevia increased incidence in such cases.

For simplicity and succinctness to the numerous data we had, we decided on the one hand to try to provide all the available data for future use from the rest of the scientific community and on the other to avoid calculated data (p-values) that we did not use because we did not consider making sense to compare. We acknowledge -as you well pointed out that hemoglobin levels should have p-values stated and have been added.

As mentioned, our starting assumption regarding increased hemorrhage associated with an active infection in placenta previa was not proven. Albeit we identified that increased hemorrhage is related to special cases of placenta previa, namely PAS and complications arising from this condition (hemostatic hysterectomies) and were able to associate an increased incidence of PAS with 1st trimester SARS-CoV-2 infection while pointing out that our secondary outcomes (prematurity, fetal outcome, and overall evolution) were not influenced according to our data and by and large fit the data already existing in the literature.

According to our data presented it appears that maternal outcome is influenced by an active SARS-CoV-2 infection and its severity during labour but we chose not to elaborate further since the specific group C did not present significant hemorrhagic differences in comparison to our control group A and did not wish to wade away far from our initial intention.

We believe that our work provides a solid stepping point to pursue a prospective study regarding PAS placenta previa and SARS-CoV-2 infection timeliness and investigate further the effect of this infection on abnormal placentation. In terms of clinical importance, we wish to draw attention to the practicing clinicians that potentially the impact of infection with SARS-CoV-2 is not limited during labour but it may impact other aspects depending on its timeliness during pregnancy.

To the best of our abilities, we searched for any available references regarding hemorrhage during pregnancy associated with SARS-CoV-2, especially with placenta previa or PAS, and to make use of it all. We acknowledge that some may have fallen below our radar and we would be ever grateful if you could specifically point out such references.

Once again, thank you for your feedback and the valuable time that you spent helping us with your review.

On behalf of all the authors,

Nikos Zygouropoulos

Reviewer 3 Report

This article is devoted to a new topic of interest, the relationship of Covid-19 and the increase in the frequency of placenta previa.

Covid-19 is a new infection that has not been fully studied, and its effect on the fetus and placenta is not clear.

This article is of interest in terms of attempting a multivariate analysis of the impact of the Sars Cov-2 virus on the mother-placenta-fetus system.

Considering that the virus enters by binding to the ACE2 receptor, which is present on many cells in the body, including the placenta, I believe that the virus can change the formation of the placenta, as well as the adhesive properties of cells, which can lead to the formation of placenta previa.

The study was built correctly, the comparison of groups is carried out in accordance with the gestational age.

Comments should include Table 1: Demographic, obstetrical and related characteristics of SARS-COV2 infection. It is not clear what is compared with what, what r refers to. It is recommended to make a note in the table.Statistical differences should be presented more clearly.

Author Response

Dear Reviewer,

Thank you for your effort and time spent reviewing our manuscript. We greatly appreciate your insights and have tried to improve our manuscript in line with your recommendations.

We have elaborated further -in text- regarding the p-values quoted in Table 1 to more clearly present both how and which groups are compared as per your suggestion.

Further, we have once again proofread the manuscript to perfect the English language used and requested a few native speakers in the field of obstetrics & gynecology but otherwise unrelated to this study to “go once over our manuscript” and point out any inappropriate or incorrect use of the language. We trust that it will meet your standard of English. In case you consider further changes are required, please do not hesitate to point these out. Without trying to be more of a burden on your precious time allocated to reviewing our manuscript, it would be of great help if you could point out the respective line number(s).

We share your belief that viral binding to the ACE-2 receptor on the placenta potential leads to or at least contributes to the appearance of placenta praevia by modifying both the placental formation and its adhesive properties. Hence we first ventured to gather evidence that the timing of the viral infection influences placentation and its relation to PAS.

We look forward to seeing further research done in this field in the future.

Once again, thank you for your constructive feedback.

On behalf of all the authors,

Nikos Zygouropoulos